# The Potential of Using an Autogenous Tendon Graft by Injecting Bone Marrow Aspirate in a Rabbit Meniscectomy Model

**DOI:** 10.3390/ijms232012458

**Published:** 2022-10-18

**Authors:** Ken Iida, Yusuke Hashimoto, Kumi Orita, Kazuya Nishino, Takuya Kinoshita, Hiroaki Nakamura

**Affiliations:** 1Department of Orthopaedic Surgery, Osaka City University Graduate School of Medicine, Abenoku, Osaka 545-8585, Japan; 2Department of Orthopaedic Surgery, Osaka Metropolitan University Graduate School of Medicine, Abenoku, Osaka 545-8585, Japan

**Keywords:** bone marrow aspirate, meniscus, autogenous tendon graft

## Abstract

Bone marrow aspirate (BMA) is an excellent source of cells and growth factors and has been used successfully for bone, cartilage, and soft-tissue healing. This study aimed to investigate the histological and biomechanical properties of autogenous tendon graft by injecting BMA and its protective effect against degenerative changes in a rabbit model of meniscal defects. Adult white rabbits were divided into untreated, tendon, and tendon + BMA groups, and meniscal defects were created in the knees. The tendon graft and articular cartilage status were evaluated by macroscopic and histological analysis at 4, 12, and 24 weeks postoperatively among the three groups. The tendon graft in the tendon and tendon + BMA groups were used for biomechanical evaluation at 4, 12, and 24 weeks postoperatively. The meniscal covering ratio in the tendon + BMA group was better than that in the tendon and untreated groups at 12 and 24 weeks postoperatively. The matrix around the central portion of cells in the tendon + BMA group was positively stained by safranin O and toluidine blue staining with metachromasia at 24 weeks. The histological score of the tendon graft in the tendon + BMA group was significantly higher than that in the untreated and tendon groups at 12 and 24 weeks postoperatively. In the tendon + BMA group, cartilage erosion was not shown at 4 weeks, developed slowly, and was better preserved at 12 and 24 weeks compared to the untreated and tendon groups. Histological scores for the articular cartilage were significantly better in the tendon + BMA group at 24 weeks. The compressive stress on the tendon graft in the tendon + BMA group was significantly higher than that in the tendon group at 12 weeks postoperatively. Transplantation of autogenous tendon grafts by injecting BMA improved the histologic score of the regenerated meniscal tissue and was more effective than the tendon and untreated group for preventing cartilage degeneration in a rabbit model of massive meniscal defects.

## 1. Introduction

The meniscus is a C-shaped fibrocartilaginous structure in the tibiofemoral region of the knee that improves congruence between the hyaline articular surfaces [1]. It is an important structure that absorbs shock, lubricates, and stabilizes the knee [2]. Meniscectomy is the most common arthroscopic surgery performed on the knee joint [3]; however, it is correlated with degenerative changes in the articular cartilage and the progression of osteoarthritis [4]. The treatment of meniscectomy-indued knee osteoarthritis remains difficult. Meniscal transplantation may be an alternative management option for certain symptomatic patients who have previously undergone total or subtotal meniscectomy [5]. Meanwhile, allografts for meniscal grafts and biologics have favorable short-term clinical and experimental results [6,7,8]. The patellar, Achilles, and semitendinosus tendons have been used as alternative treatments for meniscal defects; however, clinical outcomes have been controversial [9,10,11]. Treatment with an additional process of tendon autograft would be necessary to improve the clinical result that leads to cartilage protection.

Bone marrow aspirate (BMA) has been identified as an excellent source of cells and growth factors (e.g., PDGF, TGF-b, VEGF, bFGF) and has been used successfully for bone, cartilage, and soft-tissue healing [12]. It can release growth factors that play important roles in tissue regeneration. Liang et al. reported that bFGF and TGF-b maintained chondrogenesis of semilunar fibrochondrocytes under hypoxic conditions [13]. Moreover, the BMA fibrin clot is expected to enhance the meniscal healing of avascular tears [14]. Given these advantages, BMA may effectively promote tendon transformation into meniscus-like tissue.

This study aimed to determine the histological and biomechanical properties of autogenous tendon graft by injecting BMA and analyzing its protective effect against degenerative changes in a rabbit model of meniscal defects. We hypothesized that the autogenous tendon graft, augmented by injecting BMA, had greater histologic and biomechanical properties and exerted a protective effect on the articular cartilage than an autogenous tendon graft alone.

## 2. Results

### 2.1. Macroscopic Observation

The meniscus covering ratio was larger in the tendon + BMA (average, 0.50) and ten-don (average, 0.49) groups than in the untreated group (average, 0.13) at 4 weeks postoperatively (*p* < 0.0001 and *p* < 0.0001, respectively) (Figure 1A). At 12 and 24 weeks, the meniscus covering ratio in the tendon + BMA group (average, 0.45 and 0.41) was larger than that in the tendon (average, 0.39 and 0.34) and untreated (average, 0.14 and 0.15) groups (*p* = 0.03 and *p* < 0.0001 at 12 weeks postoperatively, respectively; *p* = 0.02 and *p* < 0.0001 at 24 weeks postoperatively, respectively) (Figure 1B). The articular cartilage was evaluated using the International Cartilage Repair Society cartilage lesion classification [15]. As for the femoral condylar cartilage, moderate cartilage erosion was observed macroscopically at 4 weeks and progressed over 12 and 24 weeks in the untreated and tendon groups. In the tendon + BMA group, cartilage erosion was not evident at 4 weeks but developed gradually; at 12 and 24 weeks, the cartilage was better preserved compared with the untreated and tendon groups (Table 1).

### 2.2. Radiology of the Tendon Grafts

No calcified changes in the anteroposterior radiographs were noted in the tendon + BMA, tendon, or untreated groups over the experimental period of 24 weeks (Figure 2).

### 2.3. Histology and Immunohistochemical Analyses of the Tendon Grafts 

Hematoxylin and eosin (H&E) staining was performed to examine histological changes in the tendon grafts. (Figure 3A) Magnification with H&E staining showed that only fibrous tissue was present in the ten-don group at 4 weeks, but fibrochondrogenesis was observed in the tendon + BMA group. At 12 and 24 weeks, oval- or round-shaped fibrochondrocytes appeared in the center of the regenerated meniscus in the tendon + BMA group. Toluidine blue staining revealed metachromasia in the three groups; metachromatic staining intensity was approximately similar in the untreated and tendon + BMA groups; however, there was much less metachromasia in the tendon group. Meanwhile, the staining was stronger in the tendon + BMA group at 12 and 24 weeks (Figure 3B). The histology score [16] was used to quantify the regeneration of the meniscus; the tendon + BMA group (average, 8.1 and 8.2) had greater regeneration than the untreated (average, 3.1 and 3.8) and tendon groups (average, 6.1 and 6.3) at 12 and 24 weeks postoperatively (*p* < 0.01 and *p* = 0.02 at 12 weeks postoperatively, respectively, *p* < 0.01 and *p* < 0.01 at 24 weeks postoperatively, respectively) (Figure 3C). Positive safranin O staining was consistently noted at the central margin of the grafts in the tendon + BMA group at 12 and 24 weeks after the surgery, which resembled the native meniscus. Positive safranin O staining was not observed in the tendon or the untreated group (Figure 4A). The amount of matrix immunostained with type II collagen increased at 24 weeks in the tendon + BMA group. (Figure 4B).

### 2.4. Evaluation of the Chondroprotective Effect of Meniscus Reconstruction

Femoral condylar cartilage degeneration progressed rapidly in the untreated group, whereas cartilage was significantly better preserved in the tendon and tendon + BMA groups (Figure 5A). The degenerative changes observed included partial fissures in the untreated group at 4 weeks, full-thickness fissures in the tendon group at 12 and 24 weeks, complete loss of cartilage in the untreated group at 12 and 24 weeks and proteoglycan washout as evidenced by decreased metachromasia with toluidine blue staining in the tendon + BMA group at 24 weeks. The Mankin scores for cartilage evaluation in the tendon (average, 1.4, 2.8, and 3.3) and tendon + BMA groups (average, 1.3, 1.9, and 2.0) were significantly lower than those in the untreated group (average, 3.3, 6.2 and 7.2) at 4, 12 and 24 weeks postoperatively (*p* < 0.01 and *p* = 0.02, at 4 weeks postoperatively, respectively, *p* < 0.01 and *p* < 0.01, at 12 weeks postoperatively, respectively, *p* < 0.01 and *p* < 0.01, at 24 weeks postoperatively, respectively) (Figure 5B). Additionally, the tibial articular cartilage was more severely degenerated than the femoral cartilage (Figure 6A). The Mankin score was significantly lower in the tendon + BMA (average, 3.3) and tendon groups (average, 4.8) than in the untreated group (average, 6.8) at 12 weeks postoperatively (*p* < 0.0001 and *p* < 0.0001, respectively). At 24 weeks, the Mankin score in the tendon + BMA group (average, 3.5) were lower than those in the tendon group (average, 4.9) and untreated group (average, 8.2) (*p* = 0.02 and *p* < 0.0001, at 24 weeks postoperatively, respectively) (Figure 6B). 

### 2.5. Biomechanical Testing 

The compressive stress of the regenerated meniscus in the tendon + BMA group (average, 4.0) was significantly higher than that in the tendon group (average, 2.2) at 12 weeks postoperatively (*p* = 0.03) (Figure 7).

## 3. Discussion

In this study, transplantation of an autogenous tendon graft by injecting BMA transformed into meniscus-like tissue. The histological score of the tendon grafts was significantly higher in the tendon + BMA group than that in the untreated and tendon groups postoperatively. The compressive stress of the tendon grafts in the tendon + BMA group was significantly higher than that in the tendon group at 12 weeks postoperatively. Moreover, the tendon + BMA group was more effective than the tendon and untreated group in preventing cartilage degeneration in a rabbit model of massive meniscal defects. 

Tendon grafts have been used in partial or total meniscus reconstruction and show different degrees of cartilage protective effects and graft remodeling outcomes due to their distinct properties and operation procedures. Tendon autografts have been used in animal studies to investigate the chondroprotective effects of meniscectomy [9]. A study using a portion of the quadriceps tendon as a meniscal autograft reported favorable results in terms of both healing and cartilage protection. However, in 2000, Johnson and Feagin conducted a pilot study wherein tendon autografts were used for lateral meniscal transplants that revealed that there was no clinical improvement or preservation of the joint space after lateral meniscal autograft transplantation, and the patient demonstrated loss of the lateral space of the joint at the time of surgery, so the patients were more likely to undergo knee replacement [11]. Numerous reports have shown that Mesenchymal stem cells (MSCs) transplantation promotes meniscus regeneration in animal models [17,18], and they reported that tissue from MSCs can enhance the regeneration of injured tissue and that the meniscus delays the progression of OA. However, it is cost- and labor-intensive to cultivate and propagate cells such as MSCs or articular chondrocytes, and two surgeries are required. 

The clinical use of autologous BMA is simple and can be prepared in the operating room as a single-step procedure. BMA has been identified as an excellent source of cells and growth factors and has been used successfully for bone, cartilage, and soft-tissue healing in single-stage surgical procedures [13,19]. In this study, the meniscus-covering ratio in the tendon + BMA group was larger than that in the tendon and untreated groups, and type II collagen in the matrix increased at 12 and 24 weeks in the tendon + BMA group. The histological score was used to quantify the regeneration of the meniscus; the tendon + BMA group had greater regeneration than the untreated and tendon groups. Additionally, percutaneous BMA injections for delayed unions or nonunions resulted in long bone bridging in >70% of patients [20]. Fortier et al. reported that delivery of concentrated BMA can result in the healing of acute full-thickness cartilage defects, which is superior to that after microfracture alone in an equine model [19]. Hashimoto et al. reported an improvement in postoperative clinical results after meniscal repair using a BMA clot for isolated avascular meniscal injury [14]. Given these advantages of a BMA, it is expected to promote the transformation of a tendon into meniscus-like tissue more effectively.

Biomechanical characteristics are important for protecting articular cartilage for meniscal regeneration. We found that the compressive stress of the tendon grafts in the tendon + BMA group was significantly higher than that in the tendon group at 12 weeks postoperatively. This result could be explained by the histological and immunohistochemical analyses of the grafted tendon. A previous study reported increased collagen content and fibroblasts in response to increased compressive stresses [21]. Otsuki et al. reported that the instantaneous compressive stress at 50% strain was approximately 5 MPa for meniscal scaffolds [22]. At 24 weeks, instantaneous compressive stress at 50% strain in the regenerated meniscus in the tendon + BMA group and tendon group was almost 5 MPa, suggesting that the autogenous tendon graft can be used as an alternative material for a meniscal scaffold. 

Our histological findings showed that chondroprotective effects on articular cartilage could be achieved with intra-substance BMA augmentation. Moderate cartilage erosion was observed at 4 weeks, which progressed over 12 and 24 weeks in the untreated group. Mankin scores throughout the literature ranged from 3.0 to 9.0 out of 14 for other in vivo meniscus tissue engineering studies, with damage often progressing over time [23]. At 24 weeks, the articular cartilage and subchondral bone were better preserved in the tendon + BMA group, suggesting that the tendon + BMA group was more effective in preventing cartilage degeneration in a rabbit model of massive meniscal defects.

### Limitations

Our study has some limitations. First, the rabbit meniscus has greater spontaneous healing potential than the human meniscus; therefore, interspecies differences must be considered [24]. Second, it is unclear how the BMA evolves in this setting. The BMA was not characterized in any way with respect to types and the number of cells or types and amounts of growth factors. Hence, it was not possible to determine whether the regenerated tissue originated from the joint capsule or the autogenous tendon graft. Third, we did not conduct quantitative assessments, statistical analysis, or grading scales in our detection of type II collagen via staining.

## 4. Materials and Methods

### 4.1. Animals

Forty-five adult female Japanese white rabbits aged 12 to 20 months (weight range, 3.2–4.3 kg; Japan SLC, Shizuoka, Japan) were used in this study. The animals were housed in cages with free access to food and water in an air-conditioned environment.

### 4.2. Surgical Procedure of the Meniscus Reconstruction with an Autogenous Tendon Graft

The rabbits were anesthetized with subcutaneous injections of ketamine (50 mg/mL; Sankyo, Tokyo, Japan) and xylazine (0.2 mg/mL; Bayer, Tokyo, Japan) at a ratio of 10:3 at a dose of 1 mL/kg body weight and antibiotics (100 mg cefazolin). In 90 knees of 45 rabbits, the hair on both knee joints was shaved, washed with chlorhexidine gluconate solution, and draped under sterile conditions. The surgical procedures were performed by an orthopedic surgeon. A medial patellar incision was made to approach the knee joints, and the patella was laterally dislocated. The knee joint was exposed to resect the entire medial meniscus and harvest the semitendinosus muscle tendon. The BMA was obtained from the lateral side of the proximal tibial of the knee joint using a 15-gauge bone marrow harvest needle (Argon Medical Device) (Figure 8A). The harvested tendon was ligated using 3-0 nylon sutures to prevent the excessive spread of BMA within the tendon; two knots were made at 12 mm intervals. A 26-gauge needle attached to a 1 mL disposable syringe (TERUMO, Tokyo, Japan) containing the BMA (100 µL) was introduced through the distal end of the ligature and advanced toward the knot at the proximal end (Figure 8B). The BMA was evenly injected into the center of the graft by pulling the needle back. Thereafter, the knot was removed, and the BMA-injected tendon was sutured to the posterior cruciate ligament and the anterior capsule with 5-0 nylon sutures. The margin of the tendon was sutured to the stump of the original medial meniscus by using multiple sutures (Figure 8C) [25]. The rabbits were allowed to walk freely in their cages after the surgery.

### 4.3. Study Design

The animals were then divided into the untreated, tendon, and tendon + BMA groups. In the untreated group, no treatment was performed on the meniscus defect after resecting the entire medial meniscus. In the tendon group, only transplantation of the semitendinosus tendon was performed. In the tendon + BMA group, transplantation of the semitendinosus tendon with a BMA injection was performed. To evaluate the changes and chondroprotective effect of these grafts, the rabbits were sacrificed at 4, 12, and 24 weeks after surgery, (*n* = 6 at each time point), and the macroscopic and histological findings among the three groups were compared. Meanwhile, biomechanical comparisons between the tendon and tendon + BMA groups were performed at 4, 12, and 24 weeks postoperatively (*n* = 6 at each time point). The study protocol is shown in Figure 9.

### 4.4. Macroscopic Observation

The tibial plateau with the tendon graft was carefully separated from the femoral condyle. Macroscopic pictures of the tendon graft, femoral condyle, and tibial plateaus were taken, and the size of the tendon graft was quantified by measuring the area ratio of the medial meniscus to the entire area of the medial tibial plateau using ImageJ software (version 1.46, National Institutes of Health, Bethesda, MD, USA) by measuring the area ratio of the medial meniscus to the entire area of the medial tibial plateau, as previously reported [26]. The medial femoral condyles and tibial plateaus of the group were harvested for subsequent analysis. In the untreated group, only a small amount of synovium filled the space of the meniscal defect.

### 4.5. Radiological Examination

All harvested regenerated meniscal tissues, including the femoral condyles and tibial plateaus, were radiographed using a soft X-ray apparatus (SOFRON; Sofron Co., Ltd., Tokyo, Japan) to detect ectopic calcified masses in the knee joint.

### 4.6. Histological Evaluation

Tendon grafts tissues, femoral condyles, and tibial plateaus were fixed in 4% paraformaldehyde for 24 h. Osteochondral specimens were decalcified in Morse’s solution (FUJIFILM Wako, Osaka, Japan) for 14 days and embedded in paraffin blocks. The specimens were then sectioned into 5 mm slices radially for the meniscus and sagittally for the center of the medial femoral condyle and medial tibial plateau and stained with hematoxylin–eosin, toluidine blue, and safranin O. Histological analysis of the tendon grafts was performed using histology scores based on the modified Pauli score on a scale of 0–15 points [16]. The Pauli score evaluated the tissue surface characteristics, cellularity, matrix, collagen fiber organization, and safranin-O–fast green matrix staining intensity [27]; the revised histology score excluded the surface of the inner border and changed safranin-O–fast green matrix staining to toluidine blue matrix staining from the original method [15]. Cartilage degeneration of the femoral condyle and tibial plateau was evaluated using the Mankin score (0–14 points) [28].

### 4.7. Immunohistochemical Examination

Type II collagen staining was performed to confirm the presence of type II collagen in the tendon grafts in the untreated tendon and tendon + BMA groups. Paraffin-embedded sections were deparaffinized with xylene and dehydrated with graded alcohols. Slides were pretreated with citrate buffer (Target Retrieval Solution [S1699], 10; DAKO Japan, Tokyo, Japan) in phosphate-buffered saline solution (PBS) for 20 min at 90 °C for optimal antigen retrieval. Endogenous peroxidases were quenched using 1.0% hydrogen peroxide in methanol for 30 min at room temperature. The slides were then rinsed with PBS and incubated with 10% goat serum for 1 h at room temperature. Thereafter, slides were incubated with anti-type II collagen primary antibody (1:200 dilution; rabbit IgG fraction; Kyowa Pharma Chemical, Takaoka, Japan) for 1 h at room temperature. After thorough washing with PBS, the slides were incubated with a peroxidase-labeled antibody (Histofine Simple Stain; Nichirei Biosciences, Tokyo, Japan) for 1 h at room temperature. After extensive washing with PBS, immunoreaction was visualized by incubating the sections for 5 min in 3,30-diaminobenzidine (Histofine Simple DAB solution; Nichirei Biosciences).

### 4.8. Biomechanical Analyses

The tendon grafts were separated from the tibia, collected, and immediately placed in a freezer at −80 °C until analysis. Before biomechanical testing, the samples were thawed, and cylindrical specimens were prepared using a 2-mm-diameter disposable biopsy punch (Kai Industries, Tokyo, Japan). They were harvested from the tendon + BMA group and compared with the tendon group to assess aggregate modulus. Before mechanical testing, each sample was cut coplanar. The heights of the specimens were measured using a digital caliper (Mitutoyo, Kanagawa, Japan). The meniscus was firmly fixed to the plate using glue and kept moist. Analyses were performed on a material testing machine (EZ Graph; Shimadzu, Kyoto, Japan) with a 100-N load cell. After the application of the preload (0.05 N), the specimens were dynamically compressed with a constant load rate of 1 mm/min up to a half-specimen height (50% strain) [22,29]. The compressive modulus from the linear ramp region and the instantaneous stress generated immediately upon reaching 50% strain values were calculated.

### 4.9. Statistical Analysis

Statistical differences between groups were calculated using a one-way analysis of variance with a post hoc Student’s t-test. *p* < 0.05 was considered significant. EZR software (version 1.38; Saitama Medical Center, Jichi Medical University, Saitama, Japan) was used for all analyses. Data are presented as mean ± SD. 

## 5. Conclusions

Transplantation of autogenous tendon grafts by injecting BMA improved the histologic score of the regenerated meniscal tissue and was more effective than the tendon and untreated group for preventing cartilage degeneration in a rabbit model of massive meniscal defects.

## Figures and Tables

**Figure 1 ijms-23-12458-f001:**
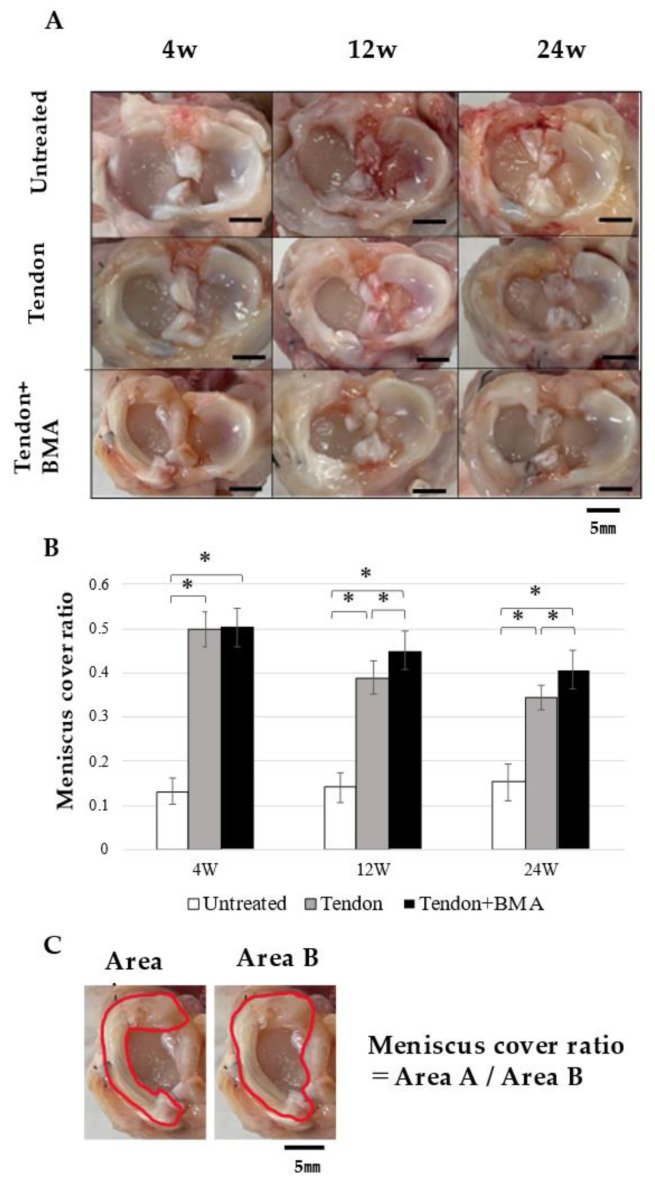
Macroscopic analyses for regenerated meniscus. (**A**) Macroscopic observation. Scale bar = 5 mm. (**B**) Meniscus cover ratio. Bars show the mean ± SD (*n* = 6). *, *p* < 0.05; At 12 and 24 weeks, the meniscus cover ratios in the tendon + BMA group were larger than those in the tendon group and untreated group. (**C**) Explanation for meniscus cover ratio, defined as the ratio of medial meniscus area (red line of Area A) to medial plateau area (red line of Area B).

**Figure 2 ijms-23-12458-f002:**
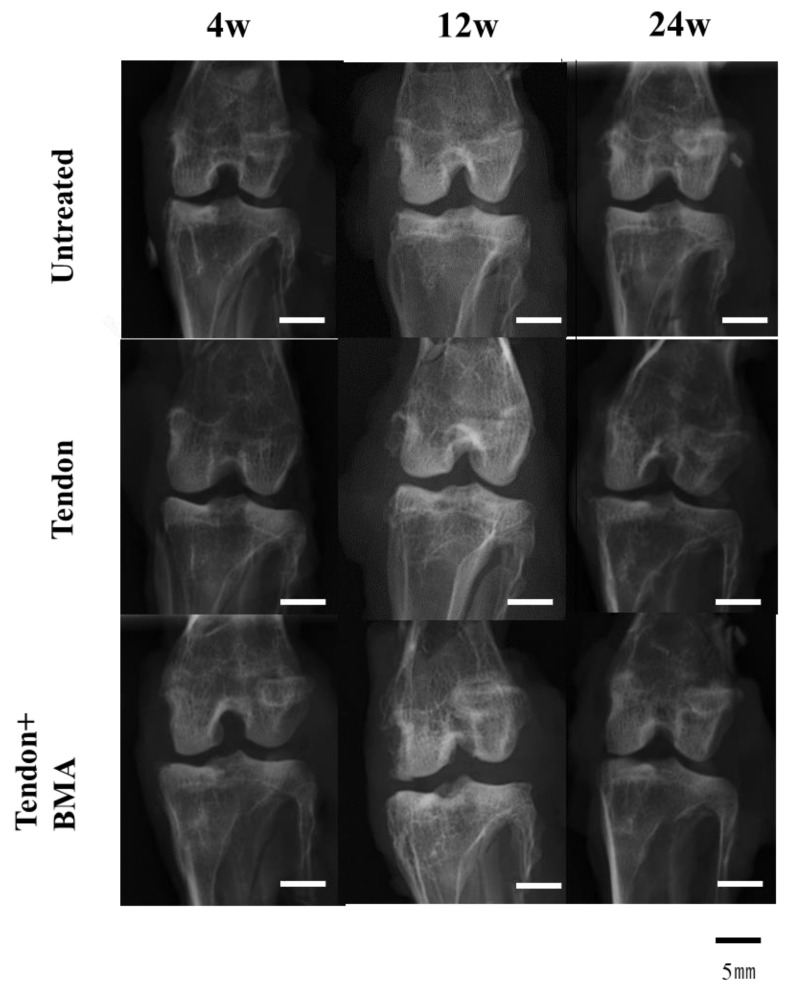
Radiographic analysis of the regenerated meniscus. No calcified change in the anteroposterior radiographs was noted in the tendon + BMA, tendon, or untreated groups over the experimental period of 24 weeks.

**Figure 3 ijms-23-12458-f003:**
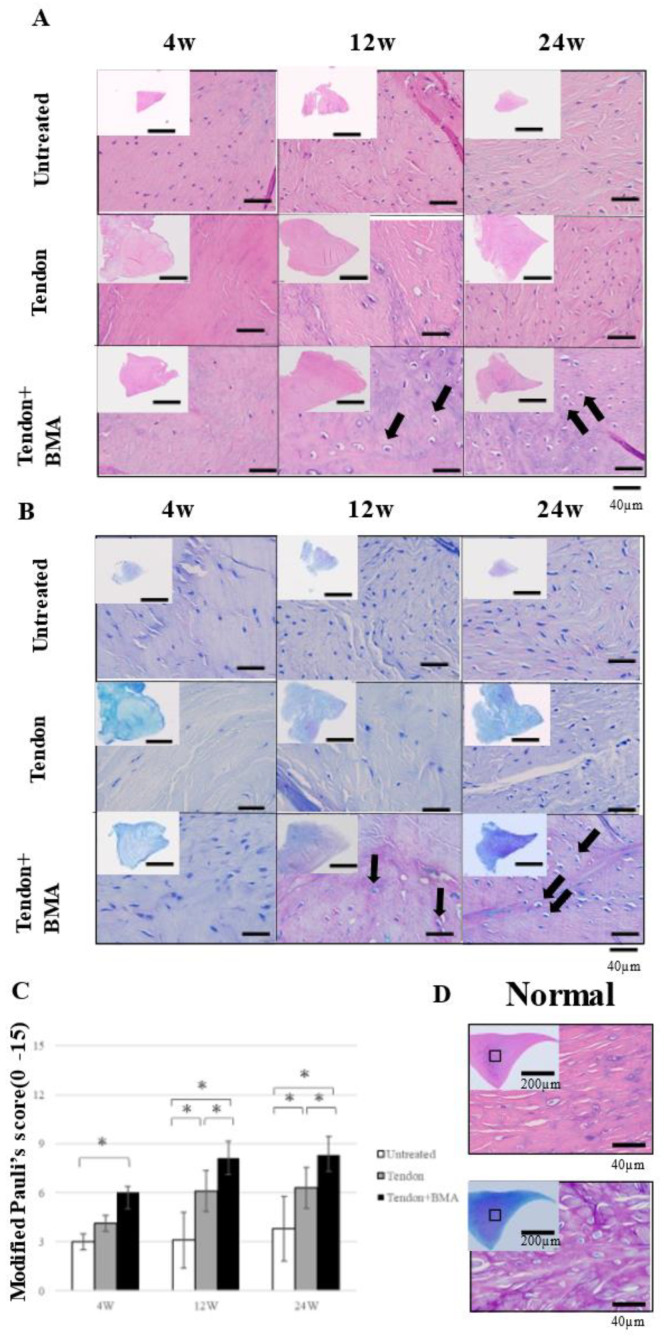
Hematoxylin and eosin (H&E) staining, toluidine blue staining, and modified Pauli’s scoring of the native meniscus and regenerated meniscus at 4, 12, and 24 weeks postoperative of three groups. (**A**) H&E staining of the regenerated meniscus revealed oval- or round-shaped fibrochondrocytes in the center of the regenerated meniscus in the tendon + BMA group. (black arrows). (**B**) Toluidine blue staining revealed metachromasia in the tendon + BMA group at 12 and 24 weeks postoperatively (black arrows). (**C**) Modified Pauli’s scoring; the tendon + BMA group had greater regeneration than the untreated and tendon groups at 12 and 24 weeks postoperatively (*n* = 6, * *p* < 0.05). (**D**) H&E staining and toluidine blue staining of the native meniscus.

**Figure 4 ijms-23-12458-f004:**
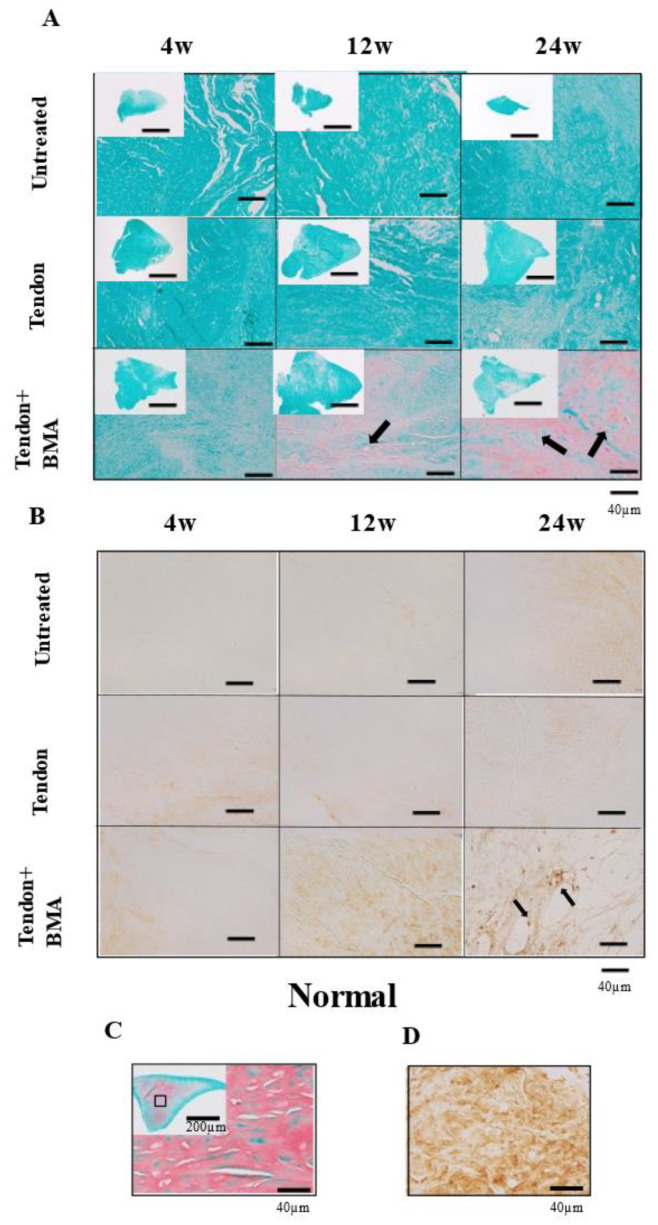
Safranin O staining and type II collagen immunostaining of native and regenerated menisci in the three groups at 4, 12, and 24 weeks postoperatively. (**A**) Safranin O staining. Positive safranin O staining was consistently noted at the central margin of the grafts in the tendon + BMA group at 12 and 24 weeks postoperatively (black arrows). (**B**) Type II collagen immunostaining. Immunohistochemistry exhibited positive staining of type II collagen in the cells at 24 weeks postoperatively (black arrows). (**C**,**D**) Safranin O staining and type II collagen immunostaining of the native meniscus.

**Figure 5 ijms-23-12458-f005:**
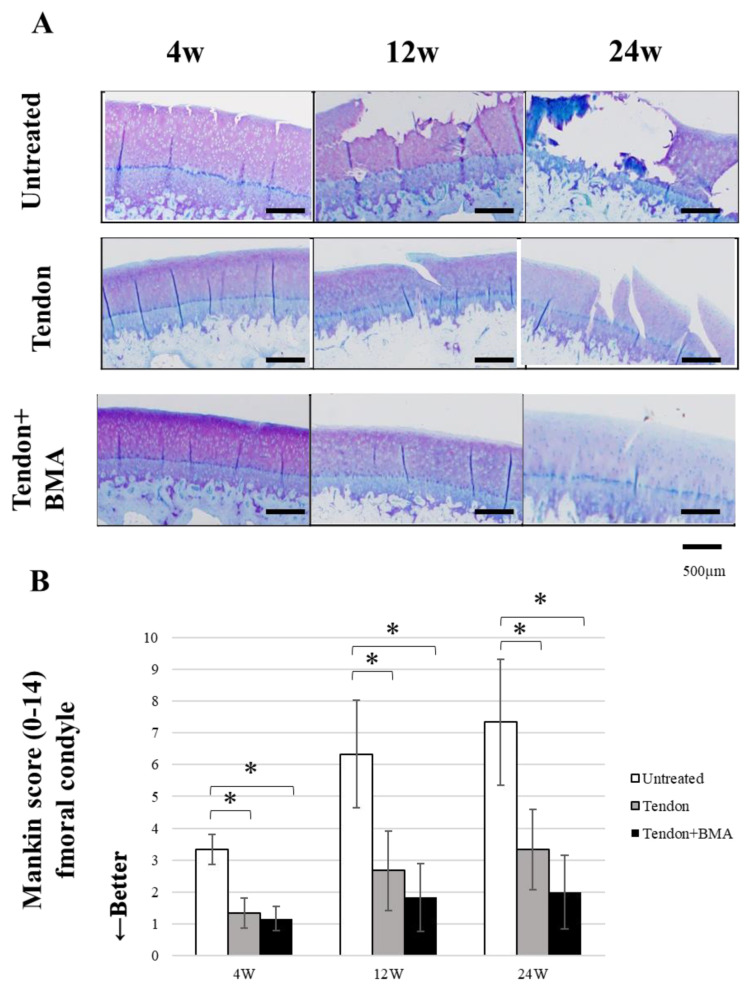
Histological analyses for articular cartilage at the medial femoral condyle. (**A**) Toluidine blue staining of medial femoral condylar cartilage. (**B**) Mankin score. (*n* = 6, * *p* < 0.05). The Mankin score in the tendon and tendon + BMA groups was significantly lower than that in the untreated group.

**Figure 6 ijms-23-12458-f006:**
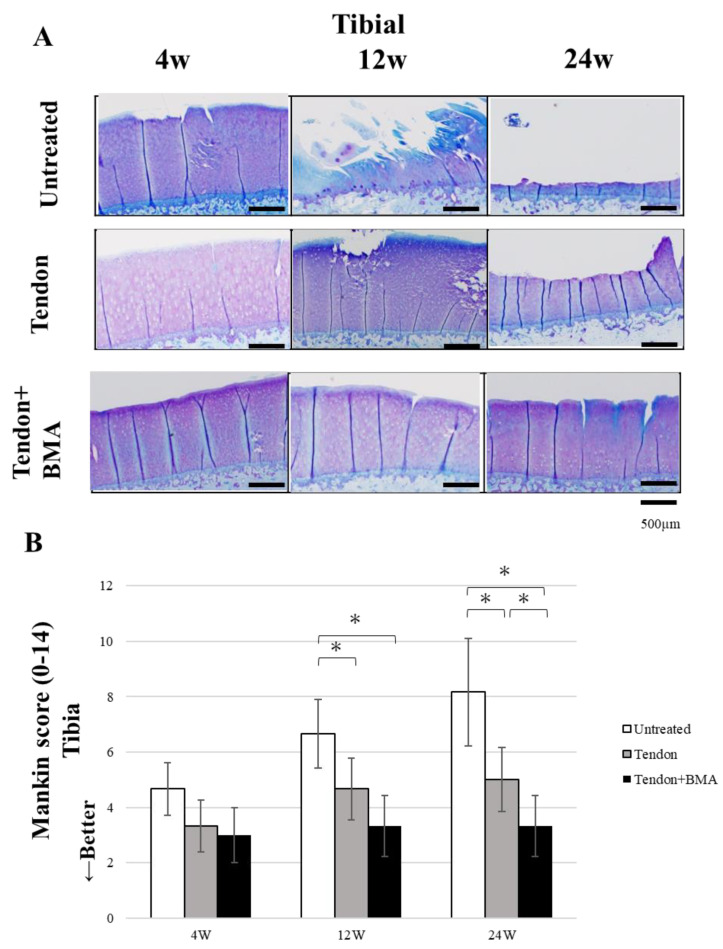
Histological analyses for articular cartilage at the medial tibial plateau. (**A**) Toluidine blue staining of medial tibial plateau cartilage. (**B**) Mankin score. (*n* = 6, * *p* < 0.05). The Mankin score in the tendon + BMA groups was significantly lower than that in the untreated and tendon groups at 24 weeks postoperatively.

**Figure 7 ijms-23-12458-f007:**
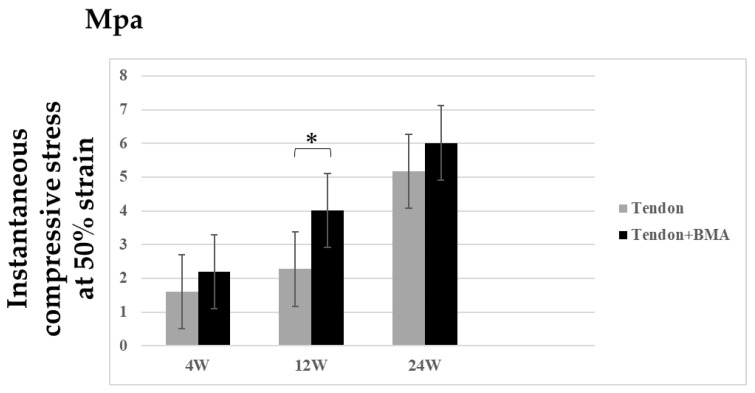
Biomechanical analysis of regenerated meniscus at 4, 12, and 24 weeks post-transplantation in tendon and tendon + BMA groups (*n* = 6, * *p* < 0.05). Values are presented as mean ± SD. The compressive stress of the regenerated meniscus in the tendon + BMA group was significantly higher than that in the tendon group at 12 weeks postoperatively.

**Figure 8 ijms-23-12458-f008:**
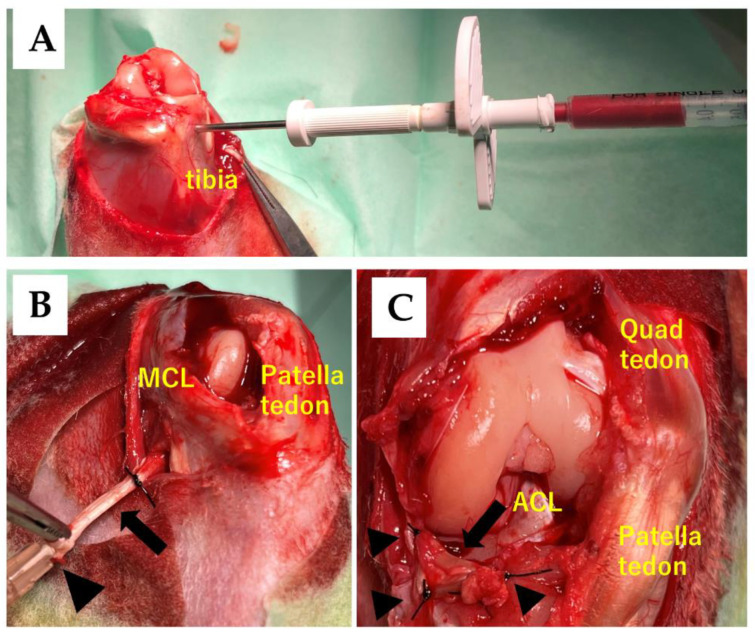
Method of injection into the semitendinosus tendon and the replacement of the meniscus tissue. (**A**) Bone marrow aspirate (BMA) was obtained from the lateral side of the tibial of the knee joint using a 15-gauge Bone Marrow Harvest Needle. (**B**) BMA was injected using a 26-gauge needle (black arrowhead) that was inserted into the center of the rabbit’s semitendinosus tendon (black arrow). Abbreviations: MCL, medial collateral ligament. (**C**) The tendon graft was sutured using 5-0 nylon sutures (black arrowhead) to the posterior cruciate ligament (PCL), the posteromedial corner of the tibia, the anteromedial capsule, and the anterior stump of the original medial meniscus. Abbreviations: ACL; anterior cruciate ligament.

**Figure 9 ijms-23-12458-f009:**
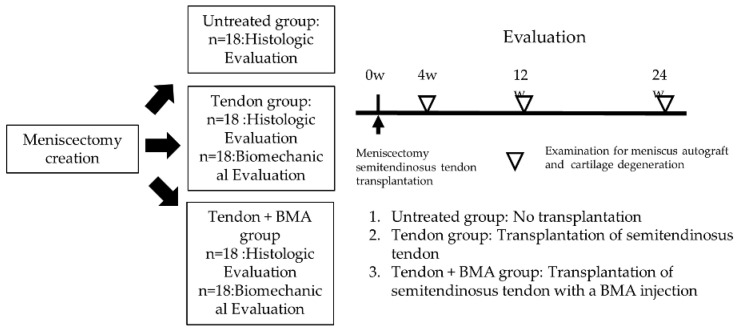
Study protocol. Rabbits were sacrificed at 4, 12, and 24 weeks after surgery (*n* = 6 at each time point), and the macroscopic and histological findings among the three groups were compared. Meanwhile, biomechanical comparisons between the tendon and tendon + BMA groups were performed at 4, 12, and 24 weeks postoperatively (*n* = 6 at each time point).

**Table 1 ijms-23-12458-t001:** Macroscopic scores based on ICRS scoring system [15].

	4 Weeks			12 Weeks			24 Weeks		
	Scores	*p* Value		Scores	*p* Value		Scores	*p* Value	
Untreated group	0.83 ± 0.37	0.54 0.69	0.16	1.83 ± 0.37	0.04 0.21	0.11	2.0 ± 0.57	0.03 0.04	0.01
Tendon group	0.5 ± 0.5			1.0 ± 0.58			1.16 ± 0.37		
Tendon + BMA group	0.33 ± 0.47			0.33 ± 0.47			0.5 ± 0.5		

Note. Data are presented as mean ± standard deviation. Data expressed as p values with significance at <0.05.

## Data Availability

Not applicable.

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
