# Peer review of "The Potential of Using an Autogenous Tendon Graft by Injecting Bone Marrow Aspirate in a Rabbit Meniscectomy Model"

_ijms, 2022, doi:10.3390/ijms232012458_

Round 1
Reviewer 1 Report
This manuscript describes an in vivo study conducted to determine the potential benefit of bone marrow aspirate augmentation on a tendon autograft used for meniscal replacement. The experiment, which was performed in rabbits, included appropriate untreated and tendon-only controls. Basing success on several outcomes, including macroscopic appearance, semi-quantitative histology, immunohistochemistry, and biomechanical properties, is considered a strength of the study. The main weaknesses are biased reporting of the results and overstatement of the advantage of BMA over tendon alone. More complete and objective description of results is needed, as well as tempering of conclusions so that they are supported by results. Specific problematic issues, a few of which are major, are enumerated below.
1. Lines 57-58: Reword this for clarity; for example, “We hypothesized that the autogenous tendon graft, augmented by injecting BMA, had greater histologic and biomechanical properties…”
2. Lines 73-75: Reword to clarify that it is the cartilage, and not its erosion, which is better preserved.
3. A two-way ANOVA, with factors of time and treatment, could have been run on quantitative data, which would have allowed for the examination of the main effect of treatment. Posthoc tests could then be used to analyze specific pairwise differences.
4. Lines 92-93: Representative 24-week radiographs would greatly bolster reader confidence in this observation.
5. Fig. 2: Explain the potential source of the regenerated meniscus.
6. Fig. 2A: Where are the cell nuclei in 4W tendon? The tissue looks essentially acellular.
7. Lines 97-98: There is no obvious increase in cellularity of tendon+BMA as compared to tendon, which is presumably the evidence of cell proliferation.
8. Lines 98-100: At the resolution of images provided, fibrochondrocytes cannot be identified by morphology that is distinct from other cells.
9. Lines 100-101: In the images provided in Fig 2B, metachromatic staining intensity is about the same in the untreated and tendon+BMA sections; there is much less metachromasia in the tendon section.
10. Figure 2 legend should indicate the 0-15 scale for the modified Pauli scoring system.
11. Lines 108-109: Should this be “tendon or the untreated group”? There should not be tendon associated with the untreated group.
12. Lines 109-111: Relative to what? The only positive staining this reviewer observes is in the tendon+BMA group at 24 weeks, indicating that collagen type II production increased with postop duration, becoming detectable at 24 weeks.
13. Figure 4A: All the images seem quite out of focus (even the originals), which makes them difficult to interpret.
14. Lines 170-184: Results should include a description of the degenerative changes observed, including complete loss of cartilage, partial and full-thickness fissures, and proteoglycan washout as evidenced by decreased metachromasia with toluidine blue staining.
15. Figures 4 and 5A: It is worth noting the prominent vertical lines which are probably folds in the sections.
16. Figure 6: Vertical axis label should be <Instantaneous/Equilibrium> compressive stress at 50% strain.”
17. Line 252-254: This statement is false. There was substantial cartilage degeneration in all experimental groups. A case can be made that there was less degeneration in the tendon+BMA compared to tendon and untreated, but articular cartilage was far from normal by 12 weeks postop.
18. Line 260-261: Where were the results reported to be promising if they were not published? Personal communication?
19. Line 291-295: Was meniscus successfully regenerated using the Otsuki study? The comparison to this scaffold is meaningful only if its use resulted in substantial regeneration of meniscus-like tissue.
20. Line 293: Should be compressive stress (not force). Specify whether it was instantaneous or equilibrium stress.
21. Line 292-293: Under what strain was the stress measured? Was it the same 50% strain as in the current study?
22. Line 297-303: This paragraph does not acknowledge the apparent degenerative changes in articular cartilage which occurred in the tendon and tendon+BMA groups. While these were not as severe as those in the untreated group, they should not go unmentioned.
23. Line 305-306: A reference supporting this assertion should be cited.
24. Lines 305-310: Another limitation is that the BMA was not characterized in any way with respect to types and the number of cells or types and amounts of growth factors.
25. Line 427-429: Specimens were compressed between smooth, impermeable platens? Please clarify.
26. Line 429: Undoubtedly, stress relaxation ensued immediately after the actuator stopped moving. Described when stress was measured. Was it the instantaneous stress generated immediately upon reaching 50% strain, or was it the stress to which the sample equilibrated after a period time held at 50% strain? Calculation of the instantaneous and/or compressive modulus would be helpful for comparison to the properties of native meniscus.
27. Line 437-438: This conclusion is NOT supported by results. As mentioned above, cartilage degeneration was by no means prevented in any of the experimental groups.
Author Response
Thank you for your reviews and helpful suggestions. We hope that our responses address your concerns.
Please see the attachment.

Reviewer 2 Report
About BMA and its use for bone healing has long been known. This work focuses on its introduction into the menisci of rabbits and the study of the histological and biomechanical properties of the autogenous tendon graft. Special defects were made in animals. The condition of the tendon graft and articular cartilage was assessed by macroscopic and histological data 4, 12 and 24 weeks after surgery in three groups. I liked the work, we can see a good solid research, there is also extensive statistics. Transplantation of autologous tendon grafts through the BMA has been shown to improve the histological evaluation of regenerated meniscus tissue. I recommend the work for publication, there are a few small remarks:
1. I recommend making changes in the structure of the work and inserting section 4 between sections 1 and 2, cause it which contains a good extensive description of the medical procedures and experimental methods used. It would be more logical.
2. The position of the black arrows in Figs. 2b, 3a,b is not entirely clear. It is not clear what to pay attention to, little is written about it in the text. Describe the structures and their position in more detail.
3. I didn't quite understand why the confidence intervals showing the validity area are the same in Figs. 4b (grey and black columns, 12W and 24W), 5b (4W all, 12 W all and 24 W grey and black)?
4. I recommend expanding the conclusion, one sentence is not enough. Moreover, there is something to write.
Author Response
Thank you for your review and valuable suggestions. We hope that our responses both clarify and address your concerns.
I recommend making changes in the structure of the work and inserting section 4 between sections 1 and 2, cause it which contains a good extensive description of the medical procedures and experimental methods used. It would be more logical.
「Author response」 Thank you for your suggestion. Although we agree with your observation, the instructions of the target journal specify that the Discussion precede the Material and Methods section; hence, we cannot change the structure of the manuscript.
The position of the black arrows in Figs. 2b, 3a,b is not entirely clear. It is not clear what to pay attention to, little is written about it in the text. Describe the structures and their position in more detail.
「Author response」 Thank you for your observation.
「Author action」
Accordingly, we have replaced Figures 2B, 3A, and 3B with high-resolution images.
- I didn't quite understand why the confidence intervals showing the validity area are the same in Figs. 4b (grey and black columns, 12W and 24W), 5b (4W all, 12 W all and 24 W grey and black)?
「Author response」 Thank you for your question. The confidence intervals in the figures is as follows: Figure 5b: grey, 1.247 and black, 1.067 at 12 weeks; grey, 1.247 and black, 1.154 at 24 weeks; Figure 6b (all grey and black): 0.942, 1.247, and 1.950 at 4 weeks; 0.92, 1.105, and 1.154 at 12 weeks; 1.00, .105, and 1.154 at 24 weeks.
I recommend expanding the conclusion, one sentence is not enough. Moreover, there is something to write.
「Author response」 Thank you for your suggestion.
「Author action」
Accordingly, we have revised the Conclusion to “Transplantation of autogenous tendon grafts by injecting BMA improved the histologic score of the regenerated meniscal tissue and more effective than tendon and untreated group for preventing cartilage degeneration in a rabbit model of massive meniscal defects.” (lines 558-561)
Round 2
Reviewer 1 Report
The revised manuscript is a significant improvement. Methods and results are more thoroughly and accurately described, and limitations are acknowledged. Most importantly, the scaled-back conclusions are supported by the results.
Author Response
We thank you for your thoughtful suggestions and insights. The manuscript has benefited from these insightful suggestions. I look forward to working with you to move this manuscript closer to publication in the International Journal of Medical Sciences.